# Full-length *Plasmodium falciparum* myosin A and essential light chain PfELC structures provide new anti-malarial targets

Dihia Moussaoui[1], James P Robblee[2], Daniel Auguin[3], Elena B Krementsova[2], Silvia Haase[4], Thomas CA Blake[4], Jake Baum[4], Julien Robert-Paganin[1]*, Kathleen M Trybus[2]*, Anne Houdusse[1]*

[1]Structural Motility, Institut Curie, Paris Université Sciences et Lettres, Sorbonne Université, CNRS UMR144, Paris, France; [2]Department of Molecular Physiology and Biophysics, University of Vermont, Burlington, United States; [3]Laboratoire de Biologie des Ligneux et des Grandes Cultures (LBLGC), Université d'Orléans, INRAE, USC1328, Orléans, France; [4]Department of Life Sciences, Imperial College London, South Kensington, London, United Kingdom

*For correspondence:
julien.robert-paganin@curie.fr (JR-P);
Kathleen.Trybus@med.uvm.edu (KMT);
Anne.Houdusse@curie.fr (AH)

**Competing interests:** The authors declare that no competing interests exist.

**Abstract** Parasites from the genus Plasmodium are the causative agents of malaria. The mobility, infectivity, and ultimately pathogenesis of *Plasmodium falciparum* rely on a macromolecular complex, called the glideosome. At the core of the glideosome is an essential and divergent Myosin A motor (PfMyoA), a first order drug target against malaria. Here, we present the full-length structure of PfMyoA in two states of its motor cycle. We report novel interactions that are essential for motor priming and the mode of recognition of its two light chains (PfELC and MTIP) by two degenerate IQ motifs. Kinetic and motility assays using PfMyoA variants, along with molecular dynamics, demonstrate how specific priming and atypical sequence adaptations tune the motor's mechano-chemical properties. Supported by evidence for an essential role of the PfELC in malaria pathogenesis, these structures provide a blueprint for the design of future anti-malarials targeting both the glideosome motor and its regulatory elements.

## Introduction

Parasites from the genus *Plasmodium*, the causative agents of malaria, are responsible for half of a million deaths per year (*WHO, 2018*). Despite significant effort and money having been devoted for developing vaccines and new preventive treatments, the malaria parasites are becoming resistant to current artemisinin-based therapies (*Haldar et al., 2018*). The global death rate from malaria has recently started to rise after many years of decrease (*WHO, 2018*), emphasizing the necessity to develop new interventions, particularly because climate change may further expand the range of *Anopheles* mosquitoes (*Hertig, 2019*; *Ryan et al., 2020*).

The mosquito-borne parasite *P. falciparum*, the deadliest species that infects humans, alternates between motile and non-motile stages. Locomotion of apicomplexan parasites occurs by a process called gliding motility (reviewed in *Frénal et al., 2017*). This mode of displacement and the infectivity of the parasite rely on a macromolecular assembly called the glideosome that is anchored in an inner membrane complex located ~25 nm below the parasite plasma membrane (PPM). The core of the glideosome consists of an atypical class XIV myosin A (PfMyoA) and a divergent actin (PfAct1). PfMyoA is a short myosin with a conserved globular motor domain and a lever arm that binds two light chains: an essential light chain (PfELC) and myosin tail interacting protein (MTIP) (*Green et al.,*

*2017*; *Bookwalter et al., 2017*). The N-terminal extension of MTIP binds to integral membrane proteins called GAPs (glideosome associated proteins), which anchors the myosin so that it can interact cyclically with actin (*Jones et al., 2012*; *Figure 1a*) (reviewed in *Frénal et al., 2017*). PfMyoA is a critical molecule in the parasite life-cycle because it powers the fast motility (~2 µm.s$^{-1}$) required during motile sporozoite stages (*Münter et al., 2009*), and is essential for providing the force (up to 40 pN) needed for non-motile merozoites to invade erythrocytes (*Crick et al., 2014*; *Robert-Paganin et al., 2019*).

The molecular motor myosin has a conserved ATPase cycle in which the state of hydrolysis of the nucleotide drives both the association with actin and movements of the lever arm. Structural reorganization initiated at the active site is transmitted allosterically by connectors, and amplified by the lever arm (*Figure 1—figure supplement 1a, c and d*) (reviewed by *Robert-Paganin et al., 2020*). We recently solved the X-ray structures of the *P. falciparum* MyoA motor domain in three states of this ATPase cycle (*Robert-Paganin et al., 2019*), which revealed how the lack of consensus residues in the connectors is compensated by the presence of a unique N-terminal extension of the heavy chain (*Figure 1—figure supplement 1d*). Phosphorylation of Ser19 within this extension directly tunes speed and force production (*Robert-Paganin et al., 2019*). The motor moves actin at high-speed and exerts low ensemble force when phosphorylated; conversely, it produces more force at the expense of slower speed when unphosphorylated (*Robert-Paganin et al., 2019*; *Figure 1—figure supplement 1b*). According to this model, phosphorylated PfMyoA would allow the parasite to move at high velocity during motile stages such as the sporozoites, and when unphosphorylated, would provide the high force necessary for merozoites to enter erythrocytes in invasive stages (*Robert-Paganin et al., 2019*).

The motor domain is not the only region of PfMyoA lacking myosin consensus sequences. The lever arm typically contains 'IQ motifs' (consensus sequence IQxxxRGxxxR) that binds light chains or calmodulin. In PfMyoA, both the first IQ motif and the PfELC that binds to it are so degenerate in their sequence that the existence of an essential light chain was only recently recognized (*Green et al., 2017*; *Bookwalter et al., 2017*; *Figure 1a*). The structure of MTIP in complex with a *Plasmodium* IQ2 peptide is the only structural information available for the PfMyoA lever arm (*Bosch et al., 2006*; *Bosch et al., 2007*; *Douse et al., 2012*). Further structures are required to establish how interactions between the motor domain and the lever arm influence the overall dynamics and properties of the motor.

Here, we present the X-ray structures of full-length (FL) PfMyoA complexed to PfELC and MTIP in two states of the motor cycle. These structures, together with molecular dynamics and small-angle X-ray scattering (SAXS), reveal the specific orientation of the lever arm in the pre-powerstroke state (PPS). This atypical priming, which enables a larger powerstroke, is stabilized by the converter and PfELC forming specific interactions with the motor domain. Kinetic and motility assays on mutant PfMyoA constructs show how the specific priming and unique sequence adaptations tune the motor properties of this atypical myosin. The lever arm structure explains how the PfELC and MTIP light chains evolved to recognize degenerate IQ motifs. The PfELC, a weak link in forming a fully functional motor, is shown here to be essential for red blood cell invasion, thus providing a novel target for the design of anti-malarial compounds.

## Results

### FL structures of PfMyoA reveal a specific priming of the lever arm

We determined the crystal structures of FL PfMyoA with two light chains bound (PfMyoA/PfELC/MTIP-Δn) in the post-rigor (PfMyoA•FL-PR) and in the pre-powerstroke (PfMyoA•FL-PPS) states at 2.5 Å and 3.9 Å resolution, respectively (*Supplementary file 1a*; *Figure 1b and c*). To investigate the structural consequences of the deletion of the N-terminal extension (N-ter) of the PfMyoA heavy chain (HC), we also crystallized the truncated construct (PfMyoA-ΔNter/PfELC/MTIP-Δn). The structure was solved in the post-rigor (PR) state at 3.3 Å resolution (PfMyoA•ΔNter-PR) (*Figure 1—figure supplement 2a and b*). In all the constructs, MTIP was truncated and lacking the N-terminal extension (residues 1–60) because this extension is predicted to be intrinsically disordered. For all three structures, the electron density was well-defined for the motor domain and the lever arm, in

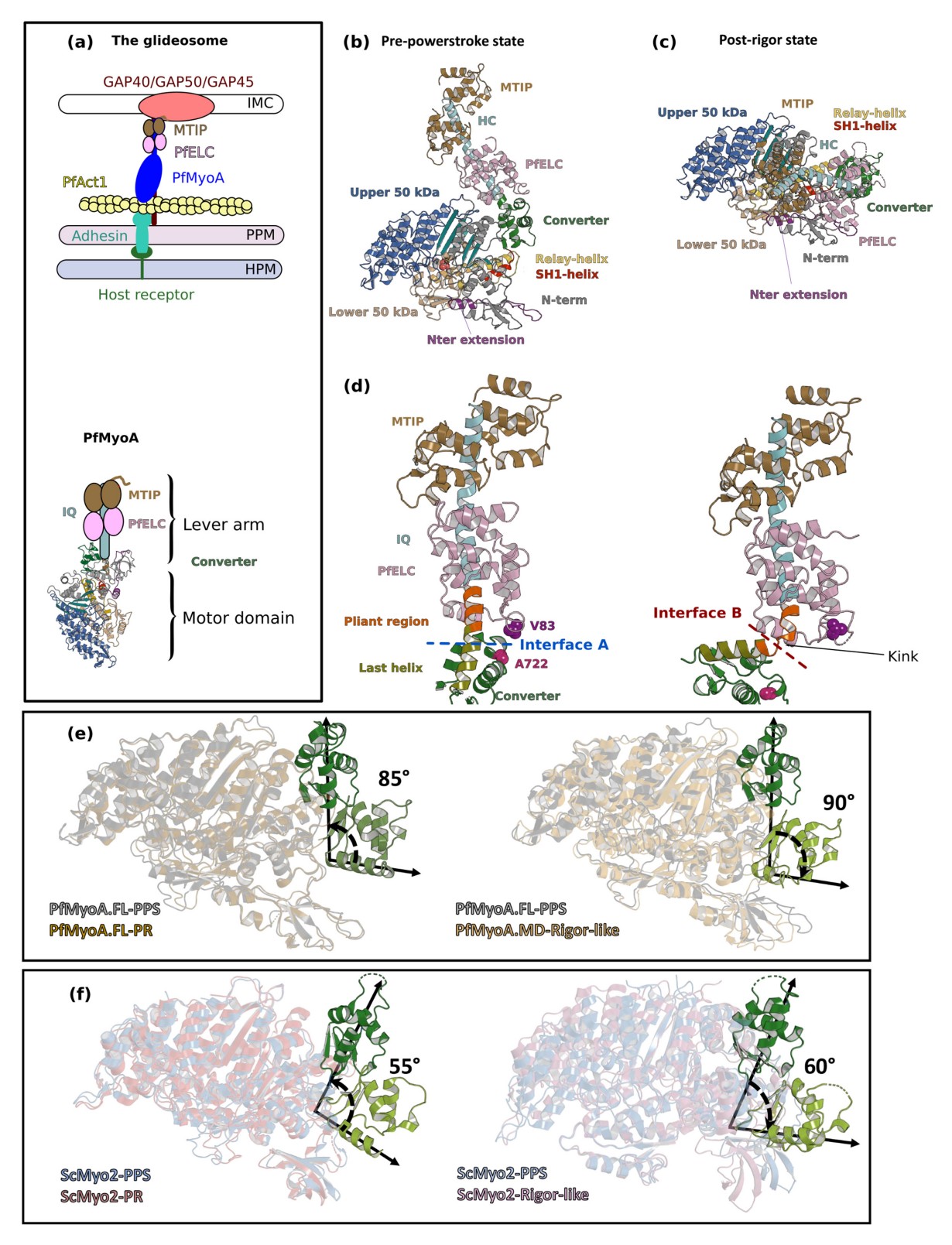

**Figure 1.** Crystal structures of PfMyoA in the Post-rigor (PR) and the Pre-Powerstroke (PPS) states. (a) (Left) The PfMyoA motor is located in the intermembrane space of the parasite. PfMyoA (blue) binds two light chains, PfELC and myosin tail interacting protein (MTIP). MTIP connects the motor to the glideosome-associated proteins (GAP) complex, which is anchored in the inner membrane complex (IMC). PfMyoA cyclically interacts with PfAct1 filaments, which are bound to adhesins from the parasite plasma membrane (PPM); these adhesins also bind receptors from the host cell plasma

*Figure 1 continued on next page*

*Figure 1 continued*

membrane (HPM). The displacement of PfAct1 filaments by PfMyoA drives parasite gliding motility. (Right) The crystal structure of the motor domain of PfMyoA has been solved (*Robert-Paganin et al., 2019*), but the lever arm structure was not known. (**b,c**) Overall structures of the full-length PfMyoA motor in the PPS and PR states, displayed so that their N-terminal subdomains adopt a similar orientation. As expected, the orientation of the converter and lever arm differs in these two states. (**d**) The lever arm has been built in these two states of the motor, revealing the structure of the two bound light chains, PfELC and MTIP, displayed here in a similar orientation. The kink in the lever arm helix at the end of the converter (pliant region in orange; last helix of the converter in deep olive green) induces different converter/PfELC interfaces in the PPS (interface A, left) compared to the PR state (interface B, right). To illustrate that the two interfaces are different, two reporter residues are displayed as spheres, A722 from the converter and V83 from PfELC. These residues are the part of interface A but not the part of interface B. (**e**) and (**f**) represents the recovery stroke (left) and the powerstroke (right) for PfMyoA and scallop myosin 2 (ScMyo2), respectively. Structures of ScMyo2 used: PR (PDB code 1S5G); PPS (PDB code 1QVI). The online version of this article includes the following figure supplement(s) for figure 1:

**Figure supplement 1.** The atypical and tunable mechanical cycle of PfMyoA.
**Figure supplement 2.** The crystal structure of the PfMyoA-PR state displays a kink at the pliant region.
**Figure supplement 3.** Electron density in the PfMyoA structures.
**Figure supplement 4.** The orientation of the lever arm of PfMyoA in PPS differs in the structures of the MD and of the FL.

particular for the interfaces between the HC and the PfELC and the interface between the two light chains (*Figure 1—figure supplement 3*).

These structures allow the description of the lever arm that was not present in the previously published PfMyoA motor domain (MD) structures (*Robert-Paganin et al., 2019*). When comparing the PR and PPS states, the overall fold of both the IQ region of the HC/PfELC/MTIP and converter regions are highly similar. Major differences are located at the ELC/converter interface when the PR and the PPS are compared (*Figure 1d*), predominantly due to a sharp kink in the pliant region (*Houdusse et al., 2000*) observed in the PR structures that promotes contacts between the ELC and the motor domain (*Figure 1—figure supplement 2c*). SAXS experiments demonstrate that this 'folded-back' position of the lever arm is not primarily populated in solution and is selected by crystal contacts. In solution, the lever arm adopts an extended conformation as seen in other myosins (*Figure 1—figure supplement 2d and e–h*).

The converter orientation is the same in both the motor domain (PfMyoA•MD-PR) (*Robert-Paganin et al., 2019*) and the FL (PfMyoA•FL-PR) post-rigor structures (*Figure 1—figure supplement 4a*). In the PPS states, crystallized with ADP and Pi analogs, however, the converter orientation is less primed by ~30° in the MD structure compared with that found in the FL structure (*Robert-Paganin et al., 2019*; *Figure 1—figure supplement 4b*). SAXS experiments validated that the PPS of PfMyoA adopts the more primed orientation in solution (*Figure 1—figure supplement 2e*). The MD structure with ADP.VO$_4$ bound thus likely corresponds to an intermediate state populated during the recovery stroke, close to that of the fully primed PPS (*Figure 1—figure supplement 4c and d*). Interestingly, the priming of the lever arm in the TgMyoA•MD-PPS structure (*Powell et al., 2018*) is identical to that of the PfMyoA•FL-PPS structure (*Figure 1—figure supplement 4e*). No major difference is found in the motor domain when the FL and MD structures are compared (*Figure 1—figure supplement 4f*). Notably, the lever arm is more primed in the PPS state of PfMyoA compared with that of conventional myosins, leading to a larger powerstroke than most myosins.

The FL structures of PfMyoA can be used to define the lever arm swing occurring during the PfMyoA motor cycle. The converter is reoriented ~90° during the powerstroke (*Figure 1e*). In conventional myosins, such as ScMyo2, the converter is reoriented only ~60° during the powerstroke (*Figure 1f*). In comparison, Myo10 undergoes a large ~140° swing during the powerstroke (*Ropars et al., 2016*) linked to both a highly primed lever arm orientation in its PPS state and an unusual rigor converter position. Specific structural features of PfMyoA thus control the lever arm repriming and larger swing amplitude during the powerstroke.

## Sequence adaptations tune distinct transitions of the cycle

Specific sequence differences in PfMyoA do not alter the global positioning of subdomains compared with conventional myosins, but instead affect the stability of structural states the motor explores and tune the kinetics of transitions between these states. Deletion of the Nter enhanced the duty-ratio (time spent strongly bound to actin) by greatly slowing ADP release to the extent that it rate-limited the overall ATPase cycle, which is ~14 fold slower than WT (*Robert-Paganin et al.,*

*2019*). The similarity of the ΔNter heavy chain and WT PfMyoA structures (rmsd of 0.415 Å on 993 atoms) (*Figure 1—figure supplement 2b*) strongly argues that the dramatic kinetic differences observed between these two constructs are due to an altered equilibrium between the states of the motor cycle (*Robert-Paganin et al., 2019*).

We previously showed that Serine 19 phosphorylation (SEP19) accelerates ADP release by stabilizing the converter in its rigor orientation through a polar interaction with $^{converter}$K764 but other residues likely modulate these motor properties. In the Rigor state, $^{Nter-extension}$E6 establishes cation-π stacking interactions with $^{Switch-2}$F476 and $^{Switch-1}$R241 from the Switch-2 and Switch-1 elements of the active site (*Robert-Paganin et al., 2019*; *Figure 2a*). The charge reversal E6R mutant was designed to disrupt the interaction seen in the rigor state, a state the motor needs to populate to release ADP. The effects of the E6R mutation are similar to that observed with mutants disrupting the polar bond between SEP19 and the converter (S19A and K764E) (*Robert-Paganin et al., 2019*):~2 fold reduced maximal actin-activated ATPase,~2 fold reduced rate of ADP release that is correlated with a reduced speed of moving actin,~3 fold increased ensemble force, and ~2.5 fold faster dissociation of acto-PfMyoA by MgATP (*Figure 2b–f*, *Supplementary file 1b*). All these results are consistent with E6 stabilizing the rigor state in WT via its interactions with Switches-1 and −2.

Two residues that are conserved in most myosins but not in PfMyoA were mutated to further investigate the role of sequence differences that are unique to PfMyoA (S691G and T586F). These residues are part of the environment between the Relay and SH1 helix, two critical connectors for the control of converter repriming and the powerstroke (*Figure 2a*). The presence of S691 at the base of the SH1 helix, which is typically a glycine in all other myosins, restrains the mobility of the SH1 helix. S691G is the only mutant to date that showed faster in vitro motility and faster ADP release than WT, implying that the mutation destabilizes the strong-ADP state (*Figure 2b–d*, *Supplementary file 1b*). The enhanced speed could be attributed to an increased flexibility of the SH1 helix or a decrease in steric hindrance that favors converter movement, resulting in faster ADP release. The maximal actin-activated ATPase activity was half that of WT and not rate-limited by ADP release (*Figure 2e*, *Supplementary file 1b*). S691 is involved in the communication between the Relay and the SH1 helices by maintaining an electrostatic bond with $^{Relay}$Q494, which is part of the sequence compensation specific to PfMyoA. This bond is formed in the PR and PPS states but must be lost during the powerstroke. The absence of this bond in the mutant could reduce actin-activated ATPase because of altered communication between the connectors that affects the kinetics of the transitions in the motor cycle, introducing a slower step most likely early in the powerstroke. The nine-fold higher basal ATPase activity in the absence of actin (*Supplementary file 1b*) indicates that the pre-powerstroke (PPS) state of the mutant is less stable than WT, while the similar rate of dissociation of acto-PfMyoA by MgATP implies that the stability of the rigor state and the transition to PR are like WT (*Supplementary file 1b*). The mutant retains the ability to undergo conformational transitions when bound to actin under strain, with a modestly enhanced ensemble force (1.4-fold) compared with WT (*Figure 2f*, *Supplementary file 1b*). The role Ser691 plays in PfMyoA is thus to stabilize the PPS state and to slow the steps that are involved in rearrangements on F-actin to control the powerstroke.

Thr586 from the Wedge was replaced by the bulkier aromatic residue Phe that is present in conventional myosins (T586F). The wedge guides the movement of the relay-helix and the SH1-helix connectors depending on L50 subdomain movements that result from events in the active site (*Figure 1—figure supplement 1c*). This mutation had relatively little effect on in vitro motility speed or on ADP-release rates compared with WT (*Figure 2b–d*, *Supplementary file 1b*), despite the fact that the bulkier Phe would likely reduce the mobility around the wedge due to steric hindrance and thus impede the communication between the active site and the lever arm during the lever arm swing. The rate-limiting step for motility is not the first transition of the powerstroke that leads to the strong-ADP state on actin, but the following ADP-release step, which is not affected by this mutation. Rearrangements near the wedge thus play a minor role for ADP release, consistent with previous structural results (*Mentes et al., 2018*). The ~40% faster transition to the PR state suggests that rigor is slightly destabilized, while the elevated basal ATPase indicates a reduced stability of the PPS state (*Supplementary file 1b*). The reduced maximal actin-activated ATPase activity of the mutant indicates that other transitions in the motor cycle have slowed (*Figure 2e*, *Supplementary file 1b*). Similar to the other two point mutants (E6R and S691G), its ability to

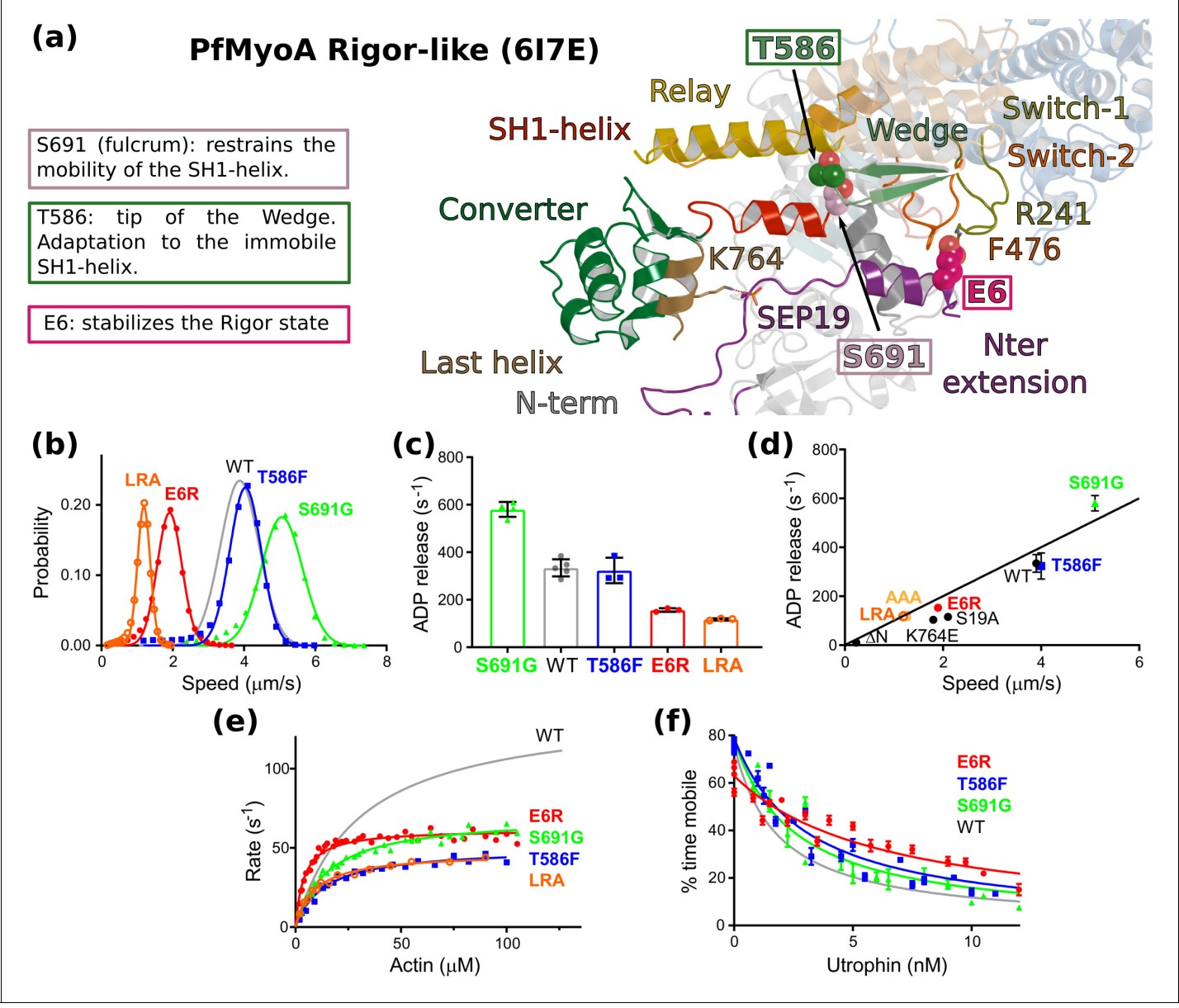

**Figure 2.** Sequence adaptations tune distinct transitions of the cycle. (**a**) Location and function of three mutated residues. (**b**) Speed distributions from representative in vitro motility assays. ***Supplementary file 1b*** shows values from additional experiments. (**c**) ADPrelease rates from acto-PfMyoA. WT data are from ***Robert-Paganin et al., 2019***. Data for LRA (three experiments, two protein preparations); E6R (three experiments, three protein preparations); T586F (three experiments, two protein preparations); S691G (four experiments, three protein preparations). Values, mean ± SD. (**d**) Correlation of ADP -release rates and in vitro motility speed. (**e**) Actin-activated ATPase activity. WT data are from ***Robert-Paganin et al., 2019*** . Data from at least two protein preparations and two experiments for each construct were fitted to the Michaelis-Menten equation. Error, SE of the fit. (**f**) Ensemble force measurements using a utrophin-based loaded in vitro motility assay. A myosin that produces more force requires higher utrophin concentrations to slow motion: E6R, 4.02 ± 0.31 nM; T586F, 2.38 ± 0.18 nM; S691G, 1.99 ± 0.19 nM; WT, 1.40 ± 0.08 nM. WT data are from ***Robert-Paganin et al., 2019***. Error, SE of the fit. Data from two protein preparations and three experiments for each mutant construct. ***Figure 6—figure supplement 1d*** shows additional data with an expanded x-axis. Skeletal actin was used for all experiments. Temperature, 30˚C. See also ***Supplementary file 1b*** for values.

The online version of this article includes the following source data for figure 2:

**Source data 1.** Source data for kinetic experiments presented in ***Figure 2***.

generate motion under force was better than WT, implying more time spent in actin-bound states of the powerstroke (*Figure 2f*, *Supplementary file 1b*). Thr586, which is an adaptation to the immobile SH1 helix, contributes to stabilizing both the PPS and Rigor states.

The FL structures are consistent with the model previously proposed for the atypical and tunable mechanism of force production (*Robert-Paganin et al., 2019*). The new mutants analyzed here validate the idea that the sequence compensations observed in the SH1-helix, the Wedge and the Nter are key-elements that not only maintain the allosteric communication between the different myosin connectors, but also tune the motor properties.

## Specific recognition of the degenerate IQ motifs in the atypical PfMyoA lever arm

PfMyoA, like other class XIV myosins, is short and lacks a tail domain (*Foth et al., 2006*). The C-terminal region of PfMyoA consists of two degenerate IQ motifs (PfIQ1 and PfIQ2) that deviate from the consensus IQ sequence (IQxxxRGxxxR) but are recognized by the native light chains, PfELC and MTIP. The recognition of PfIQ2 by MTIP has already been described (*Bosch et al., 2006*; *Bosch et al., 2007*; *Douse et al., 2012*) and involves several residues from the consensus IQ motif sequence (*Figure 3—figure supplement 1a–e*).

The FL structures allow the description of how PfELC binds to the lever arm and the interactions it forms with the motor domain. PfELC interacts with the long HC α-helix with its N-lobe in a closed conformation and its C-lobe in a semi-open conformation, as it is the case for other ELCs (*Figure 3*; *Houdusse and Cohen, 1995*). While the structure of the N-lobe is mostly conserved compared to other ELCs (*Figure 3a and b*; *Houdusse and Cohen, 1996*), the C-lobe contains an extremely short α5 helix, which consists of only one turn (α5*) followed by a short linker in which a small inserted helix turn forms (α5′*) (*Figure 3c and d*). In addition, the inter-lobe linker is shorter than canonical ELCs and forms an internal hairpin-like structure, that interacts strongly with the N-lobe and does not contact residues of the HC helix (*Figure 3e and f*). This unusual feature for an inter-lobe linker prevents the PfELC from fully surrounding the HC helix.

The sequence of the first IQ motif is adapted to the atypical features of PfELC. PfIQ1 is highly degenerate, containing none of the consensus IQ motif residues (IQxxxRGxxxR) (*Figure 4a*; *Figure 4—figure supplement 1a, b and c*). In particular, PfIQ1 lacks the glutamine and the proximal arginine that makes specific polar interactions with the C-lobe intra-lobe linker (between α6 and α7) in other myosins (*Houdusse and Cohen, 1996*; *Figure 4b*). In PfMyoA, these two residues are replaced by a valine (V781) and a glutamate (E785), respectively (*Figure 4c*). While they also bind the intra-lobe linker of the semi-open C-lobe, the change in the nature of these interactions (*Figure 4b and c*) results in a shift of the position of the intra-lobe linker relative to the HC helix (*Figure 3e and f*). Interestingly, compared to other myosins, the IQ1 motif starts one residue earlier in the PfMyoA HC sequence because a HC residue is missing in the pliant region at the end of the converter. This feature, in addition to a kink in the pliant region itself, contributes to a different orientation of the PfIQ1 HC helix (and thus the PfELC C-lobe) compared with other myosins when the converter regions are superimposed (*Figure 4—figure supplement 2*). A different converter/ELC interface is thus established (*Figure 4—figure supplement 2b and c*) and specific recognition of the semi-open C-lobe occurs both by its interaction with the converter and with the HC helix. An atypical Trp (W777) in this helix strengthens the hydrophobic interactions between PfIQ1 and PfELC (*Figure 4d and e*). Conventional myosins have a smaller side chain (I782) that interacts weakly with the C-lobe (*Figure 4d*). In PfMyoA, W777 is sandwiched between the α5*, α5′* and α6 helices, thus interacting with the hydrophobic core of the PfELC C-lobe (*Figure 4e*). These interactions between W777 and the α5* and α5′* helices are not present in the folded-back PR structure that displays a much larger kink at the pliant region and thus moves W777 away. The absence of electron density for the two helical turns of α5* and α5′* in this crystal structure indicates that lack of stabilization by the converter and the W777 side chain affects the fold of these atypical PfELC structural elements (*Figure 1—figure supplement 2c*, *Figure 4—figure supplement 1d*, *Figure 5a*). These results demonstrate how the degenerate PfIQ1 motif is adapted to the peculiar structural features of PfELC, promoting specific recognition.

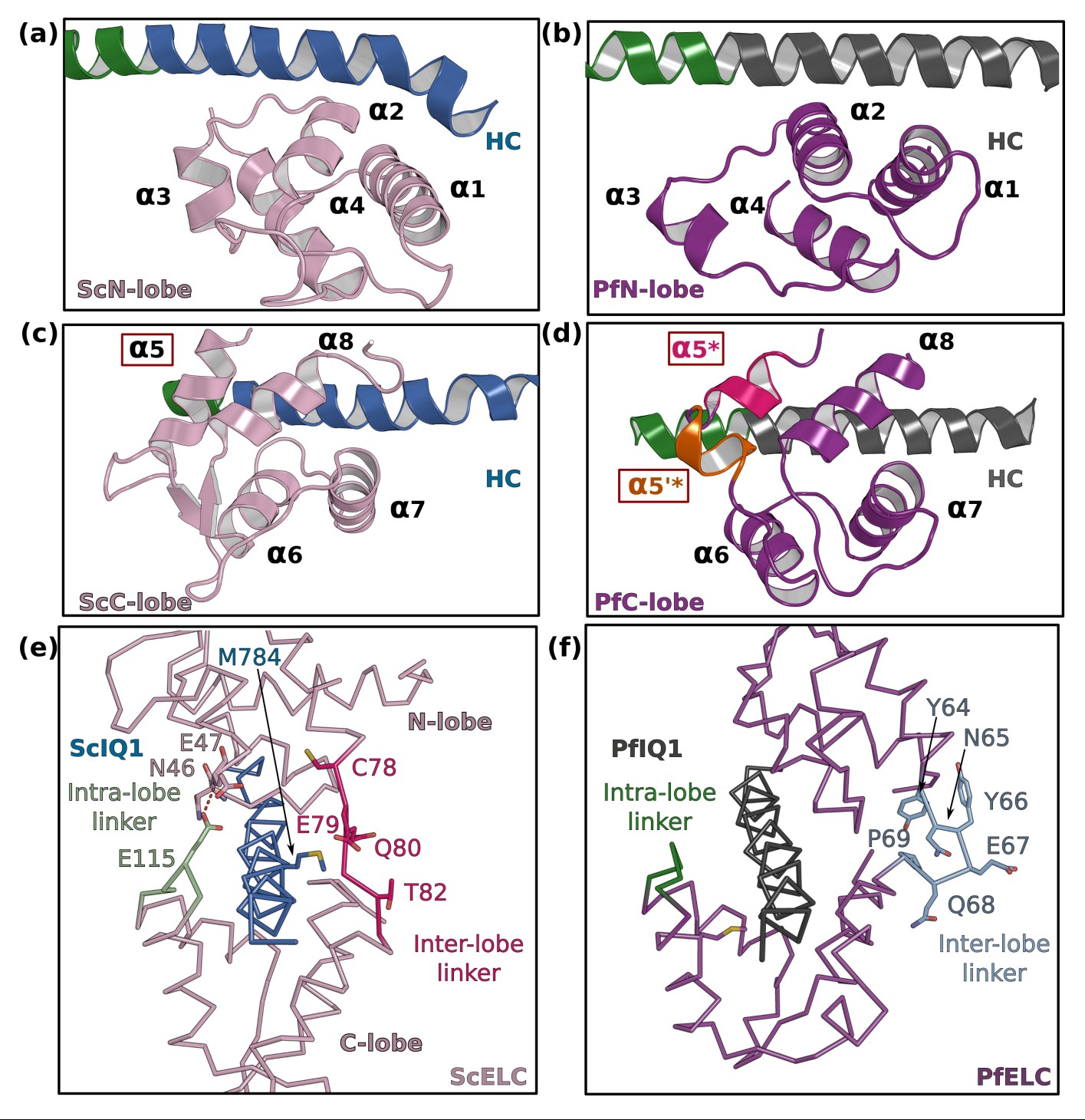

**Figure 3.** PfELC adopts an atypical fold. (**a**) The N-lobe of scallop myosin 2 ELC (ScMyo2) is compared to (**b**) the N-lobe of PfELC. The two lobes are conserved in structure, and adopt a closed conformation. In contrast, the C-lobe of ScMyo2 ELC (**c**) and PfELC (**d**) differ. The most divergent feature is the much shorter α5* helix (comprised of only one turn) and the presence of the short α5'* helix in PfMyoA. (**e**) shows the elongated inter-lobe linker of ScELC (red) compared with (**f**) the kinked, hairpin-like inter-lobe linker of PfELC (light blue). The specific structure of the inter-lobe linker makes the PfELC structure more compact than canonical ELCs and no direct interaction can occur between the lobes in PfMyoA to encircle the heavy chain, contrary to what is seen in the case of ScMyo2.

The online version of this article includes the following figure supplement(s) for figure 3:

**Figure supplement 1.** MTIP bound to PfMyoA.

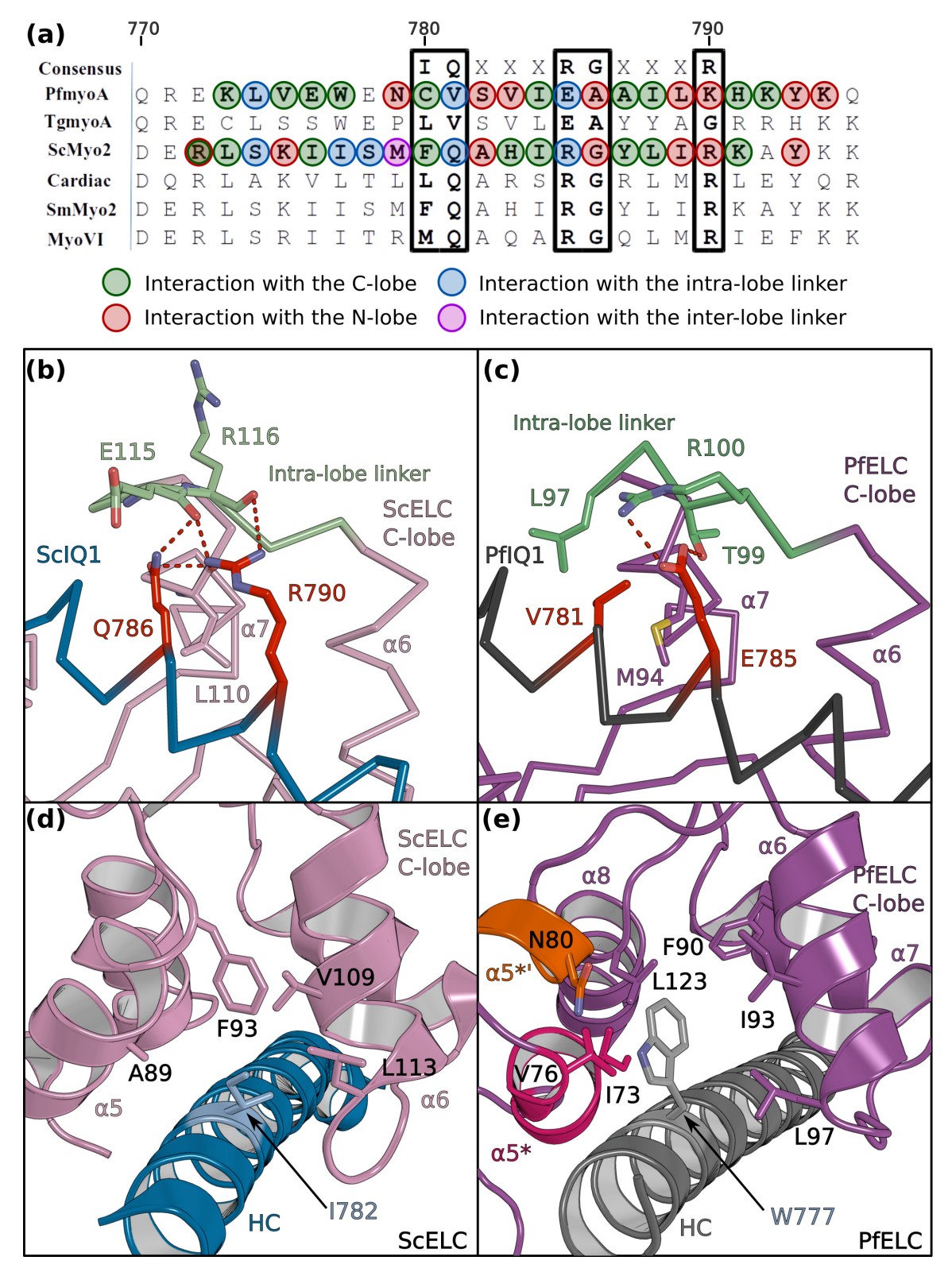

**Figure 4.** PfELC binds a degenerate IQ motif. (a) Sequence alignment of PfMyoA IQ1 to the consensus IQ motif and IQ motifs from other myosins: MyoA from *Toxoplasma gondii* (TgmyoA); bay scallop (Argopecten irradians) myosin 2 (ScMyo2); human (*Homo sapiens*) β-cardiac myosin 2 (Cardiac); human smooth muscle myosin 2 (SmMyo2); human myosin 6 (MyoVI). Consensus residues are contoured by a black box. (b) The intra-lobe linker of the scallop C-lobe interacts with the HC consensus residues with polar contacts. (c) In contrast, the intra-lobe linker of the Pf C-lobe is predominantly

*Figure 4 continued on next page*

Figure 4 continued

bound to the HC with apolar contacts. (d,e) Intra-lobe interactions maintain the semi-open C-lobe. Specificity in the recognition of the PfELC occurs via the W777 residue in PfMyoA. (d) In conventional myosins such as ScMyo2, a small side chain (I782) is found at the equivalent position, contributing to few interactions within the semi-open lobe. (e) In PfMyoA the bulky Trp (W777) is sandwiched between the α5*, α5'* and α6 helices, increasing the hydrophobic interactions with the PfELC C-lobe.

The online version of this article includes the following figure supplement(s) for figure 4:

**Figure supplement 1.** Interaction between the IQ1 motif and the ELC.

**Figure supplement 2.** The converter/ELC interface differs in PfMyoA and in ScMyo2.

## Compliance and hinge flexibility in the PfMyoA lever arm

The so-called pliant region between the converter and the lever arm (*Houdusse et al., 2000*) and the interfaces between light chains are hinges promoting lever arm flexibility (*Pylypenko and Houdusse, 2011*; *Robert-Paganin et al., 2018*). The numerous differing structural features of the PfMyoA lever arm described above change these hinges of flexibility. We describe below the converter/PfELC interface (interface A) seen in the PfMyoA•FL-PPS crystal structure, which is similar to the solution structure (*Figure 1—figure supplement 2e–h*).

The shift in the PfELC position as compared with other myosins (*Figure 4—figure supplement 2*) results in a different converter/ELC interface (*Figure 5a and b*). In canonical myosins, the converter/ELC interface is established between residues from the converter and the α5 helix and includes mainly polar and few apolar interactions (*Figure 5b*). In PfMyoA, a more rigid interface forms because it involves mainly hydrophobic contributions from the α5*, α5'*, and α6 helices whose conformations are stabilized by W777 (*Figure 5a*). The PfELC/MTIP interface is also mainly hydrophobic, involving residues from the MTIP α5 and α6 helices (H150, F151, I167, W171) and residues from the PfELC α1 helix and the following loop1 (*Figure 3—figure supplement 1f*). Interestingly, this interaction is possible only after the formation of a ~ 130° kink in the HC helix between the two IQs motifs of the PfMyoA lever arm. The kink is facilitated by proline ($^{HC}$P802) and is similar in all molecules of the asymmetric units of the structural states we crystallized (*Figure 3—figure supplement 1g*). The contacts at the MTIP/PfELC interface contribute to stabilizing the kink and providing a specific conserved feature for this rather rigid lever arm.

To further investigate the dynamics of the PfMyoA lever arm, we performed molecular dynamics with explicit solvent on the PfMyoA•FL-PPS structure, an approach successfully used to investigate other myosins (*Robert-Paganin et al., 2018*; *Robert-Paganin et al., 2019*). The first insight from this experiment is that priming of the PfMyoA lever arm is maintained throughout the time course. Even if the lever arm position slightly diverges at the beginning of the simulation, the priming is restored quickly and thus stays stable during the entire simulation (*Video 1*). As expected, the converter/PfELC and PfELC/MTIP interfaces are rigid. All the hydrophobic interactions between PfELC and MTIP are maintained during the simulation and the Converter/PfELC interface is more rigid as compared with conventional myosins (*Videos 1* and *2*). In β-cardiac myosin, the converter/ELC interface displays a controlled dynamic driven by labile polar interactions, the so-called 'musical chairs' (*Robert-Paganin et al., 2018*). These interactions between charged residues are located at the pliant region, and involve the α5 helix of the ELC and the specific top-loop of the converter (*Figure 5a and b*, *Video 2*). The dynamic associations between the musical chairs govern the flexibility of the converter/ELC interface and the conformations allowed for the top-loop (*Robert-Paganin et al., 2018*). In contrast, in PfMyoA, most of the converter/ELC interface is hydrophobic and the bulky W777 introduces rigidity at this interface. The only musical chairs of the PfMyoA lever arm are located in two regions. Charged residues from the top-loop (E725 and D726) interact alternatively with residues of the α6 helix ($^{PfELC}$D88, $^{PfELC}$N89), restricting the top-loop conformation. Other musical chairs involve residues from the converter (D720; K773 and E776) with polar residues from the α5* and α5'* helices ($^{α5*}$N75; $^{α5'*}$N77; $^{α5'*}$E78; $^{α5'*}$Q79) (*Figure 5a*; *Video 2*). We conclude that the priming seen in the PfMyoA•FL-PPS crystal structure is stable, and that the PfMyoA lever arm is more rigid than in conventional myosins.

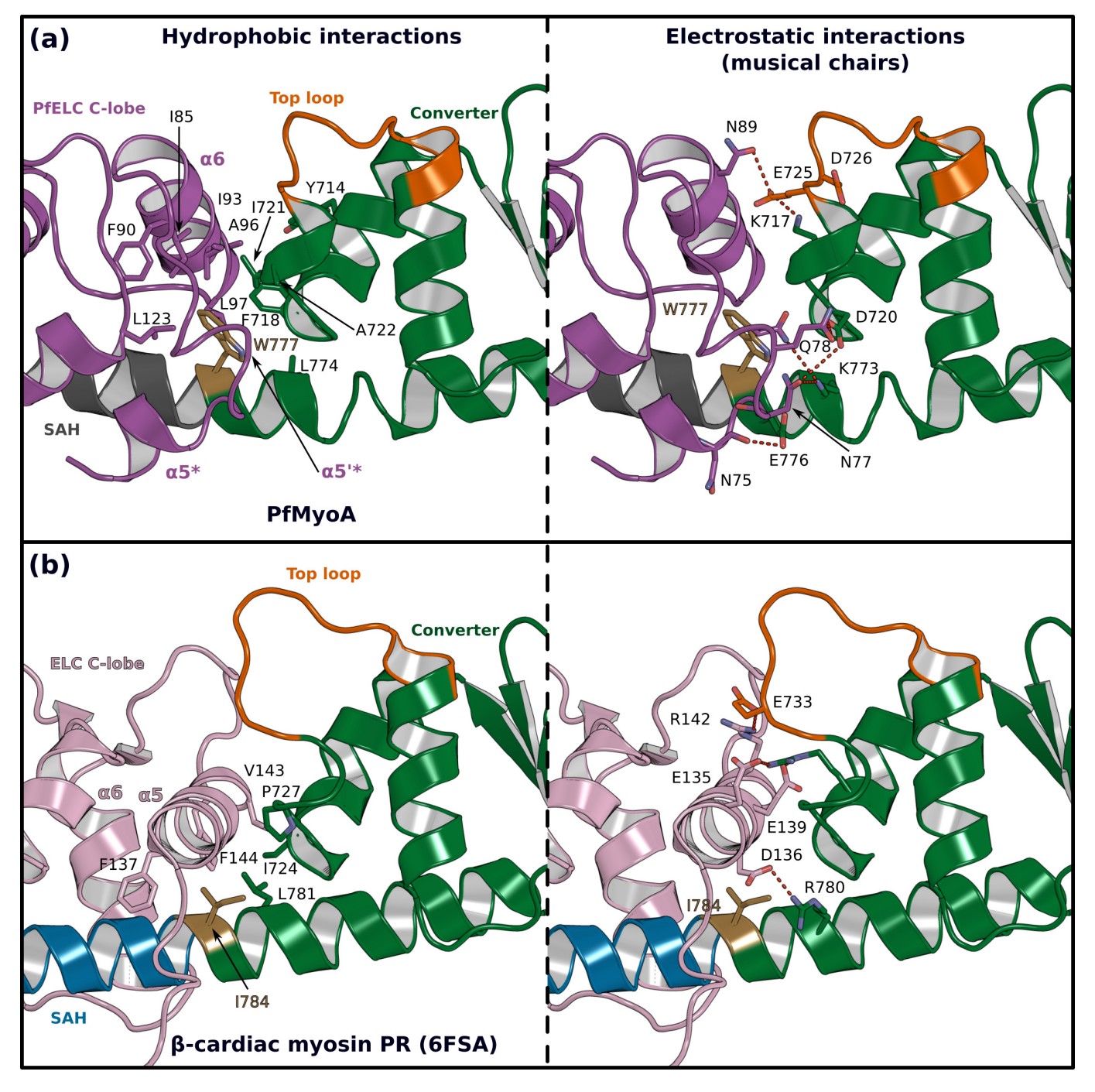

**Figure 5.** The atypical PfELC/Converter interface. Comparison of the converter/ELC interface of (**a**) PfMyoA and (**b**) β-cardiac myosin. (Left), hydrophobic contacts are displayed. (Right), polar residues responsible for the contacts identified as 'musical chairs' in molecular dynamics experiments (*Robert-Paganin et al., 2018*) are labeled.

## Motor domain/lever arm interactions stabilize the atypical PfMyoA priming

The atypical priming of the PfMyoA lever arm in the PPS state results from several interactions between the lever arm and the motor domain that are not present in conventional myosins. Conformational changes at the end of the SH1 helix are required to allow an additional Converter rotation

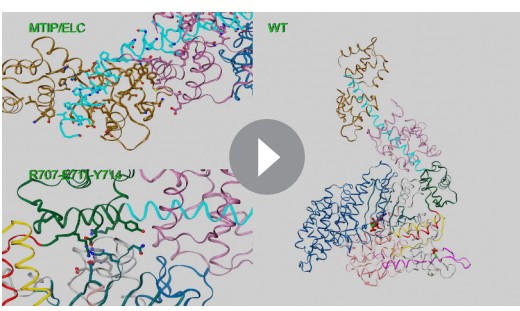

**Video 1.** Molecular dynamics performed on the structure of full-length (FL) PfMyoA in the PPS state (wild-type). Zooms show the interface between PfELC (light pink) and MTIP, and the interface between the lever arm and the motor domain.
https://elifesciences.org/articles/60581#video1

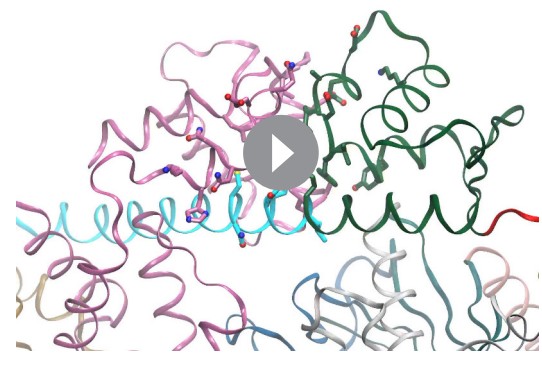

**Video 2.** Molecular dynamics performed on the structure of full-length (FL) PfMyoA in the PPS state (wild type). Zoom at the interface between the converter and PfELC showing the musical chairs.
https://elifesciences.org/articles/60581#video2

so that it can reach the PfMyoA motor domain surface with which it interacts (*Figure 1—figure supplement 4d*). These interactions involve two interfaces: interactions of PfELC with Loop-1 and the β-bulge and a specific interface between the Converter and the N-terminal subdomain (N-term) (*Figure 6a and b*). Interestingly, interactions between PfELC, loop-1 and the β-bulge are highly labile during molecular dynamics (*Video 1*), but the interface between the Converter and the N-term subdomain is highly stable and maintained throughout the time course (*Video 1*).

The converter/N-term subdomain interface can be subdivided into two regions. The first region involves a set of two polar interactions involving $^{Converter}$R707/$^{Nterm}$N177/$^{Transducer}$E660 and $^{Converter}$E711/$^{Nterm}$K183 and also hydrophobic interactions established between $^{Converter}$Y714 and $^{Nterm}$V181 and $^{Nterm}$N182 (*Figure 6b*). These interactions are maintained throughout the duration of the molecular dynamics (*Video 1*). While the position of $^{Converter}$Y714 shifts, it remains part of the interface. The second region consists in cation/π-stacking interactions between $^{Converter}$R771 from the last helix of the Converter and $^{Nterm}$H160. This interaction is also stable throughout the time course of the dynamics, while $^{Converter}$R771 occasionally establishes an interaction with $^{Nterm}$D159 (*Video 1*).

To test the hypothesis that this Converter/N-term interface is key for stabilization of the atypical priming of PfMyoA in the PPS state, we designed the triple mutant R707A/E711A/Y714A in silico. As expected, molecular dynamics show that the three mutations destabilize the Converter/Nterm subdomain interface and cause a loss of priming, demonstrating that this interface stabilizes the atypical priming of PfMyoA in the PPS state (*Figure 6c*, *Videos 3* and *4*).

The consequences of this atypical priming on the motor properties of PfMyoA were tested with two triple mutants: R707A/E711A/Y714A and R707L/E711R/Y714A (priming AAA and LRA mutants). The two sets of mutations are expected to disrupt the converter/MD interface and thus to reduce the priming of the PPS. Both triple mutants greatly impact and tune the properties of the motor similarly: they reduce the maximum actin-activated ATPase to ~30% of WT, and decrease both in vitro motility speed and the ADP release rate to a similar extent (~35% of WT) (*Figure 2b–e*, *Figure 6—figure supplement 1a–d*, *Supplementary file 1b*). The decreased ADP-release rate implies that these converter residues play a role in the stability of the strong-ADP state the motor populates when attached to actin. It is likely that a smaller step size also contributes to the reduced speed at which the mutant motor moves actin, because in silico studies show that the AAA mutant fails to maintain the WT lever arm orientation, although the Converter stays oriented in a primed orientation (*Figure 6c*).

## The PfELC is essential for *Plasmodium* invasion

As described above, the PfELC is uniquely suited to bind to the degenerate IQ motif and is also engaged in stabilizing the unusual priming of this motor, as well as being an essential element to maintain the rigidity of the lever arm. To validate this essential role and to test whether alternate

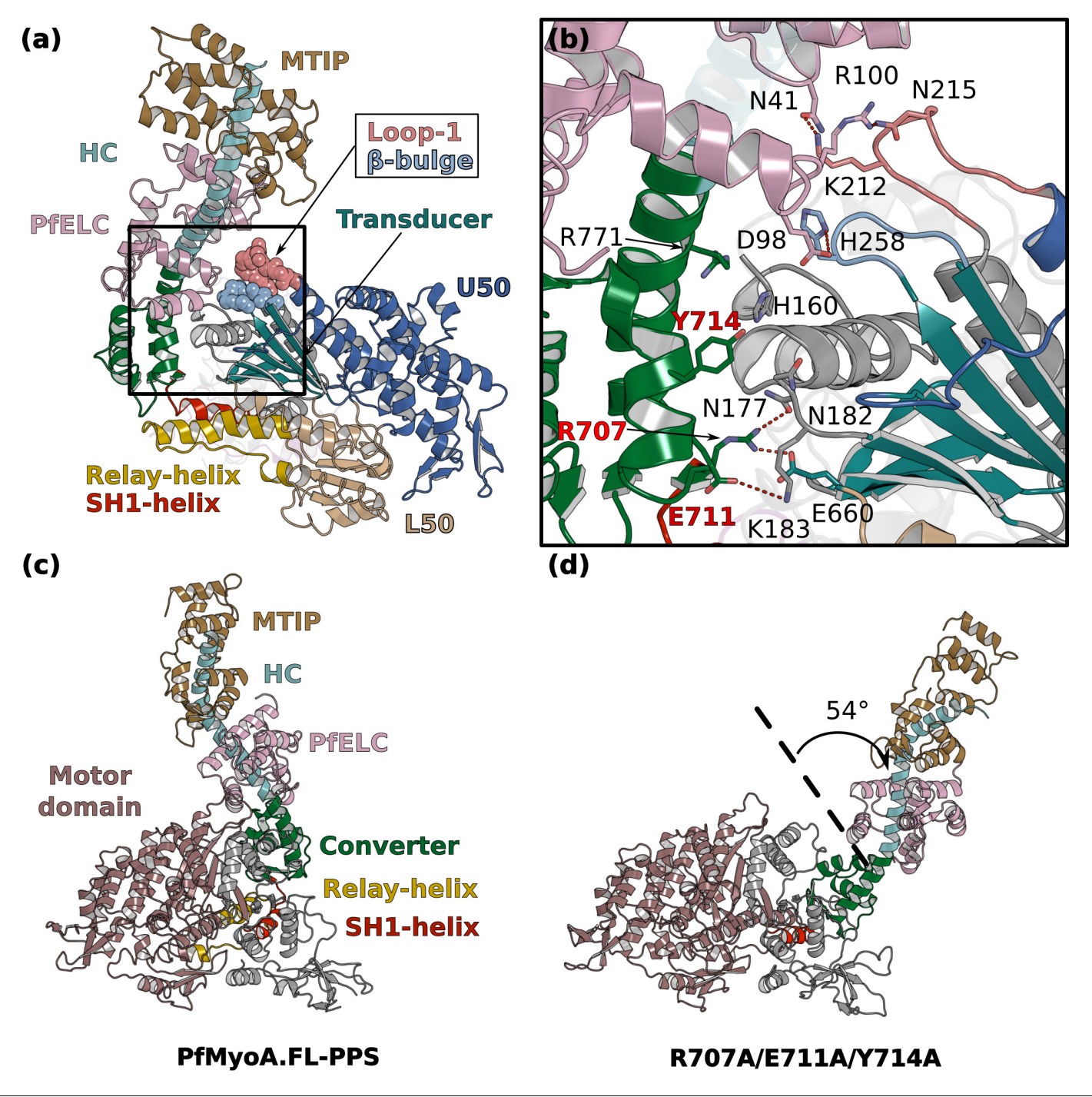

**Figure 6.** Interactions stabilizing the priming of the PfMyoA PPS state. (a) Overall structure of PfMyoA•FL-PPS. The priming of the lever arm is stabilized by interactions between elements of the lever arm and the motor domain (boxed in black). (b) Zoom on the lever arm/motor domain interface. The residues involved in the interaction are labeled. Key residues that have been mutated to disrupt the interface (see d) are labeled red. (c,d) MD simulations comparing the WT and triple mutant R707A/E711A/Y714A. (c) The primed PPS state is stable during the entire duration of the simulation (320 ns). (d) In contrast, the priming is lost with the triple R707A/E711A/Y714A mutant and the position of the lever arm deviates by up to 54°.

The online version of this article includes the following source data and figure supplement(s) for figure 6:

**Figure supplement 1.** Transient kinetics and genetic engineering of the parasite.

**Figure supplement 1—source data 1.** Source data for kinetic experiments presented in *Figure 6—figure supplement 1*.

**Figure supplement 2.** Interactions between the motor domain (MD) and the lever arm stabilizing the priming in PPS in different myosins.

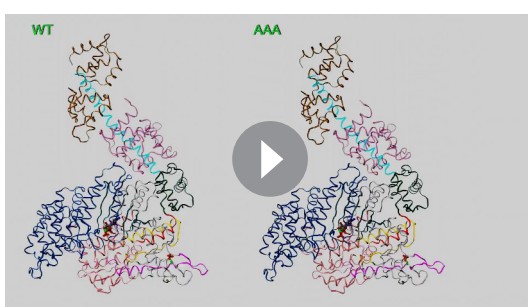

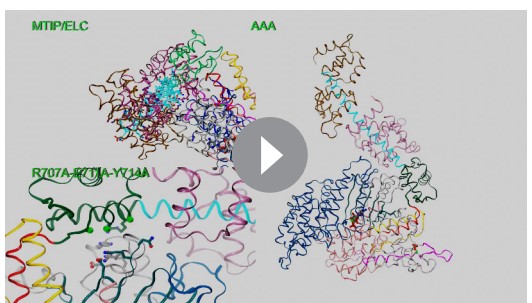

**Video 3.** Comparison of the dynamics on the structure of the molecular dynamics performed on the structures of full-length (FL) PfMyoA in the PPS state wild-type (WT) and of the in silico generated AAA mutant (R707A/E711A/Y714A).
https://elifesciences.org/articles/60581#video3

**Video 4.** Comparison of the dynamics on the structure of the molecular dynamics performed on the structures of FL PfMyoA in the PPS state wild-type (WT) and of the in silico generated AAA mutant (R707A/E711A/Y714A).
https://elifesciences.org/articles/60581#video4

light chains might compensate for PfELC function (as exists in *T. gondii* [*Williams et al., 2015*]), we generated a conditional knockout (cKO) for the *Pfelc* gene in *Plasmodium falciparum*. Using a combined process of drug selection-linked integration and engineering of artificial Cre recombinase sites into the *PfELC* gene via synthetic introns (*Jones et al., 2016*; *Birnbaum et al., 2017*; *Figure 7—figure supplement 1a*), we were able to selectively induce PfELC truncation at the protein C-terminus (lacking amino acids 93–134, predicted to render it non-functional). Integration of the introns did not affect parasite asexual growth (*Figure 7a*). Treatment with rapamycin, inducing DiCre dimerization and gene excision, led to gene truncation (as validated by PCR *Figure 7—figure supplement 1b*). Truncation was also confirmed by immunoblot and immunofluorescence assay (*Figure 7b, c and d*). Analysis of rapamycin-treated parasites was associated with a near complete ablation of red blood cell invasion over 60 hr of an asexual growth cycle, on a par with inhibition levels seen with the non-specific invasion inhibitor heparin (*Boyle et al., 2010*; *Figure 7e*). Further analysis over a single 48 hr or double 96 hr cycle showed that this inhibition of invasion was near total (*Figure 7f and g*) suggesting residual invasion after a single cycle is likely the result of incomplete gene excision. The level of invasion retardation following loss of PfELC suggests that, like the heavy chain PfMyoA (*Robert-Paganin et al., 2019*), PfELC is essential for asexual blood-stage life-cycle progression. Unlike *T. gondii,* there is no evidence for redundancy in essential light chains. Both PfMyoA motor domain and PfELC thus represent attractive targets for arresting parasite invasion in pathogenic blood stages.

## Discussion

The FL PfMyoA structures reveal how the atypical lever arm of this myosin is part of the adaptations tuning its motor cycle. This information could not be extracted from the previous PfMyoA-MD (*Robert-Paganin et al., 2019*) and TgMyoA-MD (*Powell et al., 2018*) structures that lacked the IQ region bound to its specific light chains. The PPS-FL structure of the PfMyoA motor reveals how critical adaptations in the converter and PfELC play a central role in defining and stabilizing the PPS lever arm orientation.

Unexpectedly, the lever arm of PfMyoA is ~30° more primed upon reattachment to the actin track than it is in conventional class II myosins. Designed from structural insights, in vitro and in silico experiments reveal that specific interactions stabilize the priming of PfMyoA and define specific adaptations tuning the properties of this motor. These interactions are not present in ScMyo2 (*Figure 6—figure supplement 2*). Myo10 and Myo1c are two motors for which the lever arm priming position is also large in the PPS (*Ropars et al., 2016*), (*Münnich et al., 2014*), but their lever arm orientation is stabilized by a different set of interactions when compared with PfMyoA (*Figure 6—figure supplement 2*). The contribution of the transducer in the stabilization of the primed lever arm also differs in the three myosins. It is thus tempting to speculate that the nature of these interactions will tune each motor for its specific functions. Increasing the amplitude of the lever

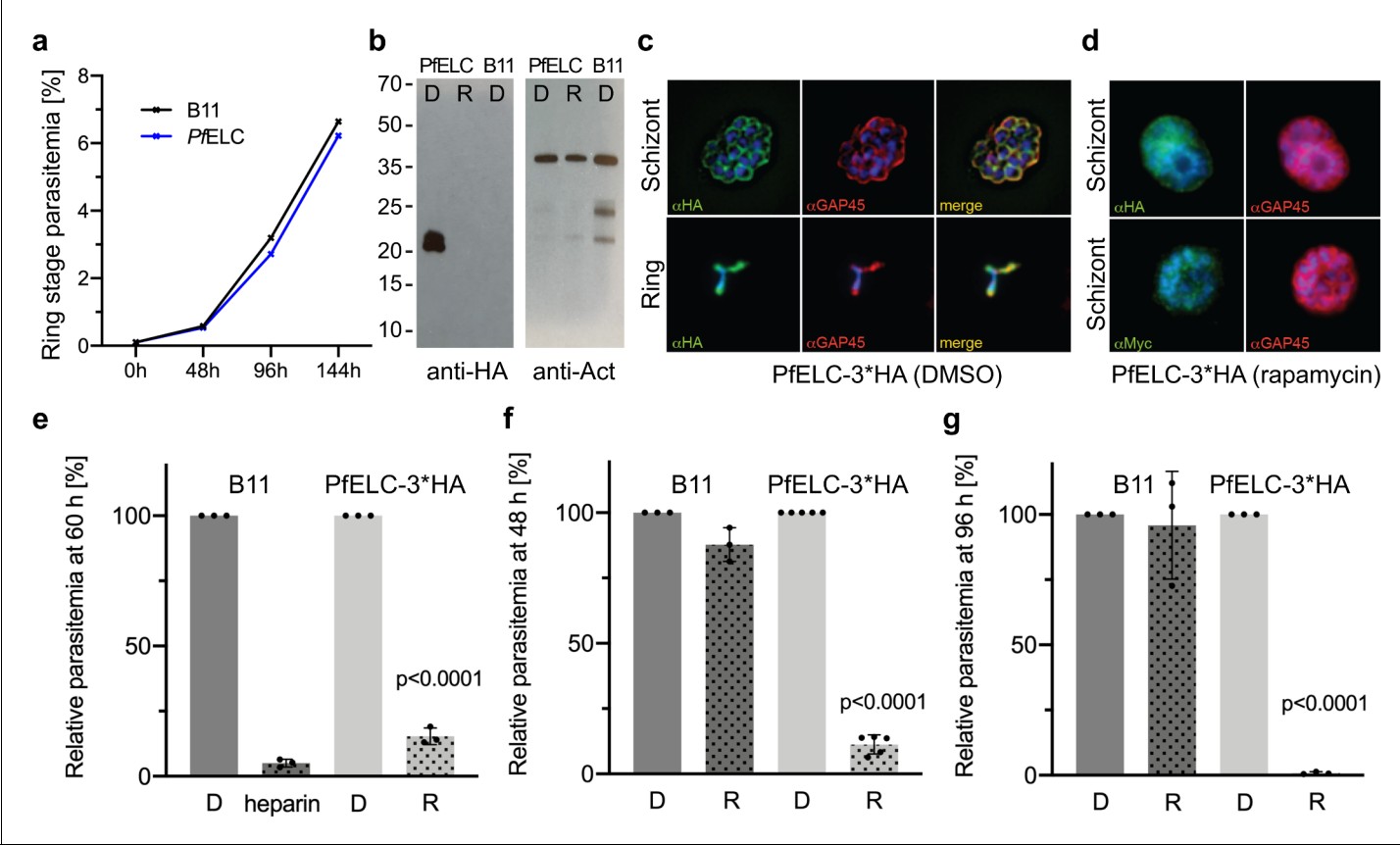

**Figure 7.** PfELC is essential for parasite invasion of the red blood cell. (a) Growth curve comparing wildtype (B11) and PfELC-3xHA parasites over the course of three cycles indicates no detrimental effect of genetic modification in the pfelc locus. Mean ring-stage parasitemia is shown of two biological replicates as determined by flow cytometry. (b) Western blot analysis of parasite extracts separated by SDS-PAGE probed with anti-HA and anti-actin antibodies as a loading control. PfELC-3xHA runs at the expected MW of ~21 kDa in mock DMSO -treated samples (D) and the HA signal is lost upon treatment with rapamycin (R), indicating successful excision. (c–d) Representative immunofluorescence assays (IFAs) of schizont and ring-stage parasites co-labeled with the IMC marker GAP45. Peripheral staining is lost after treatment with rapamycin and a diffuse/punctate pattern is seen with anti-cMyc antibodies. (e) R-treated PfELC-3xHA parasites are significantly impaired in invading red blood cells as determined by flow cytometry analysis of ring-stage parasites over 60 hr, comparable to that seen with the non-specific inhibitor heparin. A small amount of residual invasion seen after 48 hr (f) disappears after 96 hr post-treatment (g) suggesting that the absence of PfELC results in complete ablation of invasion. D-treated parasites show no invasion defect. Parasitemias were normalized to D-treated for each line, bars show mean + / - S.D.

The online version of this article includes the following source data and figure supplement(s) for figure 7:

**Source data 1.** Source data for parasitology experiments presented in *Figure 7*.

**Figure supplement 1.** Genetic integration of LoxP Cre recombinase sites into the *Pfelc* gene of *Plasmodium falciparum*.

arm priming could be a way to lengthen the step size of the motor and thus to increase the speed at which it can move actin. This is consistent with the reduced speed of in vitro motility obtained with PfMyoA mutants designed to reduce the priming of the lever arm as well as data for Myo10 indicating that this motor is tuned for performing large steps on actin bundles (*Ropars et al., 2016*). Specific interactions with the transducer may also be involved in enlarging the stroke and tuning the orientation of the powerstroke for high duty-ratio motors able to resist rearwards load, as with the Myo1b motor that anchors the cell membrane to the cytoskeleton (*Clark et al., 2005*). As it is the case for class I myosins (*Shuman et al., 2014*), (*Greenberg et al., 2015*), the duty-ratio of PfMyoA is regulated by a specific N-terminal HC extension (*Robert-Paganin et al., 2019*). The observation that converter mutations involved in priming PfMyoA also decrease the ADP-release rate suggests that these converter residues might be involved in facilitating the transitions occurring during the PfMyoA powerstroke.

Atomic structures of the PfMyoA lever arm indicate how degenerate IQ motifs of the HC are recognized by PfELC and MTIP, as well as how unusual apolar interactions occur between PfELC and the converter. These unusual features could not have been predicted from recently solved TgMyoA structures of the lever arm bound to TgELC or TgELC2 (*Pazicky et al., 2019*), because TgELC is a poor model for PfELC considering the low sequence identity (20% and 21% between PfELC and TgELC and TgELC2, respectively). Interestingly, TgELC1 and TgELC2 are homologous to classical ELCs and do not display the specific inter-lobe linker nor the atypical α5* and α5'* helices found in PfELC, which are also part of the interface with the Converter. This structural difference illustrates again that Myosin A from different organisms differ in detail due to the high divergence of the different taxa from the Apicomplexa phylum. The structure of PfELC and its specific association with the HC and MTIP also explains the cooperative binding of these elements and why PfELC does not bind to the HC in the absence of MTIP (*Bookwalter et al., 2017*). Indeed, the structures reveal a sequence shift near the pliant region resulting in a specific orientation of the HC helix to stabilize the converter/PfELC interface (*Figure 4—figure supplement 2*). The PfELC α5* and α5'* helices are involved in this interface but may not be properly folded prior to its establishment. In addition, PfELC cannot surround the IQ1 motif due to its unusual inter-lobe linker, which may contribute to diminishing a stable association with the HC. According to this hypothesis, PfMTIP may need to bind first to the IQ2 motif and its presence could add stabilizing interactions to cooperatively recruit PfELC to the PfIQ1 motif via formation of the interfaces with PfMTIP and the Converter, allowing stabilization of the PfELC α5* and α5'* helices. PfELC is thus a weak point in the assembly of the fully functional motor complex, and compounds targeting PfELC may be a good strategy for decreasing motor activity. PfMyoA lacking the ELC moves actin at half the speed of the motor containing both light chains (*Bookwalter et al., 2017*). Surprisingly, the ensemble force of PfMyoA lacking the PfELC is comparable to that of the motor with both light chains (*Figure 6—figure supplement 1e*). This is in contrast to what was seen with skeletal muscle myosin, in which the absence of the ELC results in a twofold reduction in isometric force (*VanBuren et al., 1994*). One possibility is that the atypical priming of the PfMyoA lever arm in the PPS state, which results from several interactions that are not present in conventional myosins, is responsible for allowing force to remain high in the absence of the ELC. It should be noted that video microscopy showed that PfELC cKO merozoites can still deform red blood cells, unlike cKO of PfMyoA heavy chain (*Blake et al., 2020*). Nonetheless, cKO of either PfMyoA or PfELC left merozoites entirely incapable of invasion. The reduced speed of PfMyoA in vitro without PfELC may thus be sufficient to disrupt invasion. Alternative possibilities are that in vivo other factors need to be considered that are not assessed by the in vitro assays. In vivo, the lack of the PfELC may result in proteolysis that would effectively decrease the number of functional motors. In addition, the shortened neck in the absence of the PfELC may not allow the motor to reach the actin filaments as effectively as the WT does, within the constraints of the space between the plasma membrane and inner membrane complex.

Taken together, the data presented here reveal important new findings about the PfMyoA motor, and how the lever arm is involved in tuning the specific mechano-properties of this atypical myosin. While the N-terminal HC extension of PfMyoA is involved in regulating the transition between the strong -ADP and the rigor states, the sequence adaptations located in the relay, the SH1-helix or the wedge (*Robert-Paganin et al., 2019*) influence other structural transitions essential for motor function. We show that the overall stabilization of the PPS differs in this myosin compared with all other myosins in the superfamily. Control of the recovery and powerstroke transitions differ in detail although the overall sequence allows conservation of the essential properties of a motor: the ability to couple the lever arm priming with ATP hydrolysis and control of the powerstroke via interaction with F-actin, associated with sequential controlled release of the products phosphate and ADP.

Lastly, we have also demonstrated that PfELC, like PfMyoA (*Robert-Paganin et al., 2019*), is essential in parasite asexual invasion of the red blood cell and therefore a second attractive target within the glideosome for targeting malaria parasite asexual replication, where absence of either alone or in combination would block red blood cell entry. While strategies to disrupt the glideosome have already been investigated (*Perrin et al., 2018*), inhibition of the PfMyoA motor (*Robert-Paganin et al., 2019*) or the association of the PfELC light chain opens up new strategies toward therapeutic solutions. Importantly, the structures described here provide a precise blueprint for designing just such small molecules that could prevent binding of PfELC or target PfMyoA full-length

motor activity and thus diminish glideosome function. These results inform the design of new therapies that specifically target life-cycle progression in the pathogenic blood stages of *Plasmodium*.

## Materials and methods

### Expression constructs

Full-length PfMyoA heavy chain (PlasmoDB ID PF3D7_1342600/GenBank accession number XM_001350111.1), with Sf9 cell preferred codons, was cloned into the baculovirus transfer vector pFastBac (pFB) (Thermo Fisher). A 13 amino acid linker separates the C-terminus of the PfMyoA heavy chain from an 88 amino acid segment of the *Escherichia coli* biotin carboxyl carrier protein (*Cronan, 1990*), which gets biotinylated during expression in Sf9 cells, followed by a C-terminal FLAG tag for purification via affinity chromatography. Heavy chain mutants (point mutants E6R, T586F, S691G; triple mutants R707A/E711A/Y714A and R707L/E711R/Y714A) were generated on this backbone using site directed mutagenesis. An N-terminal truncation of MTIP (MTIP-Δn, residues 1-E60 deleted and an N-terminal HIS tag), and an N-terminal truncation of the heavy chain (ΔN-terminal HC extension (ΔNter), residues 1-S20 deleted) were also cloned for use in constructs that were crystallized. Recombinant baculovirus was produced using the Bac-to-Bac Baculovirus expression system (Thermo Fisher). The mouse utrophin (NP_035812) clone was a gift from Kathleen Ruppel and James Spudich. It was modified so that utrophin residues 1-H416 were followed by C-terminal biotin and FLAG tags. It was cloned into pFastbac for production of recombinant baculovirus and subsequent expression in Sf9 cells.

### Myosin expression and purification

For biochemical characterization, FL PfMyoA heavy chain mutant constructs were co-expressed with the chaperone PUNC and the light chains (PfMTIP and PfELC) in Sf9 cells as described in *Bookwalter et al., 2017*. Two constructs were expressed for crystallization. In one, the WT heavy chain, PfELC, and MTIP-Δn were co-expressed with the chaperone PUNC. In the second, ΔNter heavy chain, PfELC, and MTIP-Δn were co-expressed with the chaperone PUNC. The cells were grown for 72 hr in medium containing 0.2 mg/ml biotin, harvested and lysed by sonication in 10 mM imidazole, pH 7.4, 0.2M NaCl, 1 mM EGTA, 5 mM MgCl$_2$, 7% (w/v) sucrose, 2 mM DTT, 0.5 mM 4-(2-aminoethyl)benzenesuflonyl fluoride, 5 μg/ml leupeptin, 2 mM MgATP. An additional 2 mM MgATP was added prior to a clarifying spin at 200,000 × g for 40 min. The supernatant was purified using FLAG-affinity chromatography (Sigma). The column was washed with 10 mM imidazole pH 7.4, 0.2M NaCl, and 1 mM EGTA and the myosin eluted from the column using the same buffer plus 0.1 mg/ml FLAG peptide. The fractions containing myosin were pooled and concentrated using an Amicon centrifugal filter device (Millipore), and dialyzed overnight against 10 mM imidazole, pH 7.4, 0.2M NaCl, 1 mM EGTA, 55% (v/v) glycerol, 1 mM DTT, and 1 μg/ml leupeptin and stored at −20°C. Utrophin purification was essentially the same as for myosin but without the MgATP steps. Skeletal muscle actin was purified from chicken skeletal muscle tissue essentially as described in *Pardee and Spudich, 1982*.

### Biochemical assays

Unloaded and loaded in vitro motility assays, actin-activated ATPase assays, and transient kinetic measurements were performed as described in *Robert-Paganin et al., 2019*. Skeletal muscle actin was used for all in vitro assays.

### Crystallization and data processing

Crystals of PfMyoA•FL-PR (10 mg.ml$^{-1}$) (Type A) were obtained at 4°C by the hanging drop vapor diffusion method from a 1:1 (v:v) of protein with 2 mM MgADP and precipitant containing 1.9M ammonium sulfate, 0.1M sodium HEPES pH 6.8, 2% PEG400. Crystals of PfMyoA•FL-PPS (Type B) were obtained at 17°C by the sitting drop vapor diffusion method from a 1:1 mixture of protein (10 mg.ml$^{-1}$) with 2 mM MgADP.VO$_4$ and precipitant containing 1M Na Citrate tribasic; 0.01M Na Borate pH 8.0. Crystals of PfMyoA•ΔNter-PR were obtained at 4°C by the hanging drop vapor diffusion method from a 1:1 mixture of protein (10 mg.ml$^{-1}$) with 2 mM MgADP and precipitant containing 2.0M Ammonium sulfate, 0.1M Sodium HEPES pH 7.5, 6% PEG400.

Crystals were transferred in the mother liquor containing 30% glycerol before and flash freezing in liquid nitrogen. X-ray diffraction data were collected at the SOLEIL synchrotron, on PX1 beamline ($\lambda$ = 0.906019 Å for type A, $\lambda$ = 0.978570 Å for type B, $\lambda$ = 0.978570 Å for type C), at 100 K. Diffraction data were processed using the XDS package (*Kabsch, 2010*) and AutoPROC (*Vonrhein et al., 2011*). Crystals type A and C belong to the $P2_12_12_1$ space group, crystals type B belong to the $P2_12_12$ space group, with one molecule per asymmetric unit for type A and C and two molecules per asymmetric unit for type B. The data collection and refinement statistics for these crystals are presented in (*Supplementary file 1a*).

## Structure determination and refinement

Molecular replacement was performed with the PfMyoA motor domain coordinates (21-768) (PDB code 6I7E for the PPS; PDB code 6I7D chain A for the PR [*Robert-Paganin et al., 2019*]) with Phaser (*McCoy et al., 2007*). The structure in the PR state (PDB code 6I7D, chain A) without ligand and water was used as a target model for type A and C crystals. PfMyoA motor domain in the PPS state (PDB code 6I7E) was used as a target model for type B crystals. Manual model building was achieved using Coot (*Emsley and Cowtan, 2004*), the structure of MTIP complexed with the cognate IQ motif peptide (PDB code 4OAM (*Douse et al., 2012*) was used to rebuild MTIP). Refinement was performed using Buster (*Bricogne, 2017*). The statistics for most favored, allowed and outlier in Ramachandran angles are for each crystal type respectively (in %): 96.40, 3.41, 0.19 for PfMyoA•FL-PR; 91.93, 6.80 and 1.27 for PfMyoA•FL-PPS; 93.12, 5.92, 0.96 for PfMyoA•ΔNter-PR.

## SAXS experiments

SAXS data were collected at the SOLEIL synchrotron, on the SWING beamline ($\lambda$ = 1.03319947498 Å). Purified PfMyoA/ELC/MTIP-Δn was extensively dialyzed against 10 mM HEPES pH 7.4, 100 mM NaCl, 1 mM DTT, 1 mM NaN3 (without any ATP) in order to remove nucleotide. We prepared two samples PfMyoA-PR and PfMyoA-PPS. Both were subsequently incubated with 2 mM MgADP for 20 min on ice, and then we added 2 mM vanadate for PfMyoA-PPS, but not for PfMyoA-PR. All samples were centrifuged at 20,000 × g for 10 min at 4°C prior to the analysis. 40 µl of the protein at 2, 5 and 9 mg.ml$^{-1}$ (17, 41 and 75 µM, respectively) were injected between two air bubbles using the auto-sampler robot. Thirty-five frames of 1.5 s exposure were averaged and buffer scattering was subtracted from the sample data. As all 2, 5 and 9 mg ml$^{-1}$ curves displayed no traces of aggregation, only the 9 mg.ml$^{-1}$ curve was used for further analysis because of the higher signal/noise ratio. The theoretical SAXS curves were calculated with CRYSOL and compared based on the quality of their fits against the different experimental curves. We computed the SAXS envelopes of the two samples with GASBOR, two programs from the ATSAS suite (*Franke et al., 2017*). The analysis and the fit of the computed SAXS envelopes and the X-ray structures were performed with FoXS (*Schneidman-Duhovny et al., 2016*). Furthermore, to analyze if we have a mixed population of PR-Closed and PR-open conformations of PfMyoA-PR in solution we used Oligomer (*Franke et al., 2017*). The model of PR-open conformation used in this study was generated by Swiss model (*Waterhouse et al., 2018*) after manual positioning of the lever arm, as found in the PPS structure in which the pliant region is not kinked.

## Molecular dynamics

Molecular dynamics experiments were performed with a procedure close to *Robert-Paganin et al., 2018*. All the systems (WT + mutants) were built with the CHARM-GUI (*Brooks et al., 2009*; *Jo et al., 2017*), with the Solution Builder module. The entire proteins were relaxed in a box containing explicit water (TIP3P) and salt (150 mM KCl) at 310.15 K in the CHARMM36m force field (*Huang et al., 2017*). The duration of the simulations was 320 ns in GROMACS (version 2018.3) (*Abraham et al., 2015*).

## Generation of the PfELC$_{loxP}$ construct

A gene fragment (GeneArt) was synthesized comprising a 454 bp targeting sequence of the *pfelc* gene (3D7_1017500), a loxPint module replacing the second intron and the last 126 bp (codon-optimized) exon, encoding the C-terminal end of PfELC (Ile93 to Ile134) and a triple hemagglutinin tag. The fragment was cloned into the NotI/AvrII site of a modified version of the conditional KO vector

(*Tibúrcio et al., 2019*), containing the SLI elements (*Birnbaum et al., 2017*) and a cMyc/FLAG tag. The resulting plasmid was purified from *E. coli* using the Qiagen Plasmid Maxi kit.

## *P. falciparum* culture and transfection

*P. falciparum* B11 parasites expressing the DiCre recombinase (*Perrin et al., 2018*) were cultured in RPMI 1640 medium containing 0.5% w/v AlbumaxII and at 4% hematocrit using human erythrocytes (blood group 0[+]) according to standard procedures (*Trager and Jensen, 1976*). Ring-stage parasites were synchronized with 5% sorbitol (w/v) and transfected with 100 µg of purified pARL PfELC$_{loxP}$ plasmid. Transgenic parasites harbouring the episomal plasmid were initially selected with 2.5 nM WR99210 (Jacobus Pharmaceuticals) for 7 days and integration into the genomic locus was achieved by G418 treatment (400 µg/ml) for 10 days.

DiCre-mediated excision was achieved by treating synchronized ring-stage parasites with 100 nM rapamycin (Sigma) in DMSO for 14–16 hr. Additionally, parasites were mock-treated with 1% (v/v) DMSO as a negative control. Cultures were washed three times with culture media to remove rapamycin and DMSO, respectively. Treated parasites were transferred into 96-well plates at 0.1% parasitemia (in triplicate) containing fresh red blood cells at 0.1% hematocrit and were allowed to proceed to the next ring-stage cycle for FACS analysis. Samples for gDNA extraction and PCR amplification, western blot and immunofluorescence analysis were taken toward the end of the treatment cycle or subsequent cycles. Parallel assays were carried out with a negative control, B11 parasites treated with heparin throughout (Pfizer, 1:25). For these assays, parasites were treated with rapamycin and washed as above, then transferred to 48-well plates at 1% parasitaemia (in triplicate) containing fresh red blood cells at 5% hematocrit and parasitaemia was analyzed as above, at around 60 hr post-treatment.

## Genotyping, western blot analysis and immunofluorescence assays

DNA was extracted using the PureLink Genomic DNA Mini kit (Invitrogen) for genotyping by PCR using primers P1 5' CATTACTTTAATTTTTATACTACTGTTTATTTTTACAGTAC 3', P2 5' CTAATCC TATTATTT AAATATTTCATATTTTTTAAACATAGATGG 3', P3 5' GGCCAGCCACGATAGCCGCGC TGCCTCG 3', P4 5' CTTGTCGTCATCGTCTTTGTAGTCCTTGTC 3' and P5 5' CAGGAAACA GCTA TGACCATG 3' and KOD Hot Start DNA Polymerase (Millipore).

Schizonts were lysed with 0.2% saponin/PBS, washed three times with PBS (supplemented with cOmplete EDTA-free protease inhibitors, Roche) and resuspended in 1x SDS loading dye. Proteins were separated on NuPAGE Novex 4–12% Bis-Tris protein gels in MES buffer (Life Technologies) and were transferred onto nitrocellulose membranes using the iBlot system (Life Technologies). Membranes were blocked in 5% skim milk/PBS (w/v) and probed with the following antibodies diluted in 5% skim milk/PBS (w/v): anti-HA (1:4000, clone C29F4, Cell Signaling), anti-cMyc (1:100, clone 9E10, Invitrogen) and rabbit anti F-actin (*Angrisano et al., 2012*)(1:1000), Horseradish peroxidase-conjugated goat anti-rabbit and goat anti-mouse antibodies were used as secondary antibodies and diluted 1:10,000 (Jackson IR).

Late stage parasites were treated with cysteine protease inhibitor E64 (10 µM) for 4 hr prior to fixation with 4% PFA/0.0025% glutaraldehyde/PBS for 1 hr. Fixed cells were permeabilised with 0.1% TX-100/PBS for 15 min, blocked in 3% BSA and probed with anti-HA (clone 12CA5, Roche), anti-cMyc (clone 9E10, Invitrogen) and anti-GAP45 (*Baum et al., 2006*) antibodies at 1:500 in 3% BSA/PBS for 1 hr, followed by three washes in PBS. Secondary Alexa Fluor conjugated antibodies (Invitrogen) and DAPI (4',6-diamidino-2-phenylindole) were diluted at 1:4000 and incubated for 1 hr, followed by three washes in PBS. Images were acquired with an OrcaFlash4.0 CMOS camera using a Nikon Ti Microscope (Nikon Plan Apo 100 × 1.4 N.A. oil) and Z-stacks were deconvolved using the EpiDEMIC plugin with 80 iterations in Icy (*de Chaumont et al., 2012*). Images were processed in Fiji/Image J (*Schindelin et al., 2012*) Representatives of 3–5 biological replicates are shown. Significance was assessed by unpaired t-test, two tailed.

## Growth analysis by FACS

Tightly synchronized ring-stage parasites were diluted to 0.1% parasitemia (in triplicate) for growth curves and transferred into 96-well plates containing fresh red blood cells at 0.1% hematocrit. Parasites were allowed to proceed to the next ring-stage cycle, were stained with SYBR Green (1:10,000,

Sigma) for 10 min and washed three times with PBS prior to collection by flow cytometry using a BD LSRFortessa. A 100,000 cells were counted and FCS vs SSC was used to gate for red blood cells, SSC-A vs SSC-W for singlets and FSC-H vs SYBR-A was applied to gate for infected red blood cells. Flow cytometry data were analyzed in FlowJo. Biological replicates are indicated in the figures.

## Acknowledgements

We are grateful to beamline scientists of PX1 (SOLEIL synchrotron) for excellent support during data collection. We thank Margaret A Titus for critical reading of the manuscript. This work was supported by National Institutes of Health multi-PI grant AI 132378 to KMT and AH and Human Frontier Science Program grant to JB and AH (RGY0066/2016). Parasite work was funded through an Investigator Award to JB (100993/Z/13/Z) and PhD studentship to TCAB (109007/Z/15/A) both from Wellcome. The AH team is part of the Labex CelTisPhyBio:11-LBX-0038, which is part of the IDEX PSL (ANR-10-IDEX-0001–02 PSL).

## Additional information

### Funding

| Funder | Grant reference number | Author |
| --- | --- | --- |
| National Institute for Health Research | AI 132378 | Kathleen M Trybus<br>Anne Houdusse |
| Human Frontier Science Program | RGY0066/2016 | Jake Baum<br>Anne Houdusse |
| Wellcome Trust | 100993/Z/13/Z | Jake Baum |
| Wellcome Trust | 109007/Z/15/A | Thomas C A Blake |

The funders had no role in study design, data collection and interpretation, or the decision to submit the work for publication.

### Author contributions

Dihia Moussaoui, Visualization, Data curation, Formal analysis, Investigation, Methodology, Writing - original draft, Writing - review and editing; James P Robblee, Data curation, Formal analysis, Investigation, Writing - review and editing; Daniel Auguin, Data curation, Formal analysis, Validation, Investigation, Visualization, Methodology, Writing - review and editing; Elena B Krementsova, Data curation, Formal analysis, Investigation, Methodology; Silvia Haase, Data curation, Formal analysis, Investigation; Thomas CA Blake, Data curation, Formal analysis, Methodology, Writing - review and editing; Jake Baum, Formal analysis, Validation, Visualization, Methodology, Writing - review and editing; Julien Robert-Paganin, Data curation, Formal analysis, Supervision, Validation, Visualization, Methodology, Writing - original draft, Project administration, Writing - review and editing; Kathleen M Trybus, Conceptualization, Data curation, Formal analysis, Supervision, Funding acquisition, Validation, Investigation, Methodology, Project administration, Writing - review and editing; Anne Houdusse, Conceptualization, Data curation, Formal analysis, Supervision, Funding acquisition, Validation, Methodology, Writing - original draft, Project administration, Writing - review and editing, Visualization

### Author ORCIDs

Dihia Moussaoui (ID) https://orcid.org/0000-0002-1605-9619
Thomas CA Blake (ID) https://orcid.org/0000-0002-8534-0025
Jake Baum (ID) http://orcid.org/0000-0002-0275-352X
Julien Robert-Paganin (ID) https://orcid.org/0000-0001-6102-2025
Kathleen M Trybus (ID) https://orcid.org/0000-0002-5583-8500
Anne Houdusse (ID) https://orcid.org/0000-0002-8566-0336

Decision letter and Author response
Decision letter https://doi.org/10.7554/eLife.60581.sa1
Author response https://doi.org/10.7554/eLife.60581.sa2

## Additional files

### Supplementary files

• Supplementary file 1. Supplementary tables. (a) Data collection and refinement statistics (molecular replacement). (b) Kinetic and motility parameters of PfMyoA mutants. (c) Dissociation of acto-PfMyoA by MgATP at 20℃.

### Data availability

The atomic models are available in the PDB under accession numbers PDB 6YCX, 6YCY and 6YCZ for the PfMyoA•FL-PR, PfMyoA•FL-PPS and PfMyoA•ΔNter-PR, respectively.

The following datasets were generated:

| Author(s) | Year | Dataset title | Dataset URL | Database and Identifier |
|---|---|---|---|---|
| Moussaoui D, Robblee JP, Auguin D, Krementsova EB, Robert-Paganin J, Trybus KM, Houdusse A | 2020 | *Plasmodium falciparum* Myosin A full-length, post-rigor state | https://www.rcsb.org/structure/6YCY | RCSB Protein Data Bank, 6YCY |
| Moussaoui D, Robblee JP, Auguin D, Krementsova EB, Robert-Paganin J, Trybus KM, Houdusse A | 2020 | *Plasmodium falciparum* Myosin A delta-Nter, Post-Rigor state | https://www.rcsb.org/structure/6YCZ | RCSB Protein Data Bank, 6YCZ |
| Moussaoui D, Robblee JP, Auguin D, Krementsova EB, Robert-Paganin J, Trybus KM, Houdusse A | 2020 | PfMyoA•FL-PR | https://www.rcsb.org/structure/6YCX | RCSB Protein Data Bank, 6YCX |

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
