## [Decision Letter]

**Acceptance summary:**

Plasmodium parasites require a Myosin A-type motor (PfMyoA) and two unique light chains (PfELC and MTIP) to infect host cells. Here, Moussaoui et al. report three new crystal structures and associated functional studies of the PfMyoA complex and reveal unique mechanochemical properties of the parasites motor complex that are required for infection. *eLife* readers will appreciate the novel insights into malaria pathogenesis and motor protein biophysics.

**Decision letter after peer review:**

Thank you for submitting your article "Structure of Full Length Plasmodium Myosin A and light chain PfELC, dual targets against malaria parasite pathogenesis" for consideration by *eLife*. Your article has been reviewed by three peer reviewers, one of whom is a member of our Board of Reviewing Editors, and the evaluation has been overseen by Suzanne Pfeffer as the Senior Editor. The following individuals involved in review of your submission have agreed to reveal their identity: James Spudich (Reviewer #2); Roger Cooke (Reviewer #3). The Reviewing Editor has drafted this decision to help you prepare a revised submission.

Summary:

Plasmodium parasites use an essential molecular machine to invade host cells: a Myosin A-type motor (PfMyoA) in complex with two unique light chains (PfELC and MTIP). Here, Moussaoui et al. report three new crystal structures and associated functional studies of the PfMyoA complex. Specifically, they determined the structures of an N-terminally truncated pfMyoA in the post-rigor state (3.3A), as well as full-length PfMyoA with both light chains bound in the post-rigor and the pre-power stroke conformations at 2.5 Å and 3.9 Å, respectively. The authors also simulated WT versus structure-based mutant motions using MD, and fit their solution-state shapes using SAXS to identify the orientation and range-of-motion of the lever-arm in the pre-power stroke state. Using their structures as a foundation, they go on to generate mutant PfMyoA motors and compare their kinetic and motility assays with the WT complex. This analysis confirmed the altered properties of the mutants and suggested how lever arm "priming" contributes to the unique mechanochemical properties of PfMyoA. The two states of the full-length PfMyoA in complex with PfELC and MTIP further enabled the authors to determine how the "degenerate" IQ motifs of the lever arm engage PfELC and MTIP. Finally, using knockout genetic studies, the authors nicely demonstrate that PfELC is requisite for a functional glidesome and that without it, the parasite cannot invade red blood cells. Pending the resolution of mostly minor issues, this work will be of interest to *eLife* readers because of its insights into malaria pathogenesis and motor protein biophysics.

Essential revisions:

1) The authors point out that the 'folded-back' position of the lever arm in the post rigor state is not primarily populated in solution (demonstrated by SAXS experiments) and argue that this state is selected/stabilized by crystal contacts. What is the primary motive to discuss this result with a figure and again in the subsection “Specific recognition of the degenerate IQ motifs in the atypical PfMyoA Lever Arm” in terms of the contacts between W777 and the α5 helices of the PfELC. It is not clear from the manuscript how significant the 'folded-back' position for this study. If this state arises from crystal packing artifacts, are the authors arguing that this state is functionally essential but too transient to capture in solution? Please clarify.

2) Prior work (Bookwalter et al., 2017), showed that in the absence of PfELC, PfMyoA-MTIP could move actin filaments at a velocity of ~1.8 µm/s and had an ATPase Vmax of ~130 s^-1^. These data suggest that PfMyoA is still functional without PfELC. However, in this study, a conditional knockout of the Pfelc gene led to a ~complete ablation of red blood cell invasion. Please cite and discuss these data.

---

## [Author Response]

Essential revisions:1) The authors point out that the 'folded-back' position of the lever arm in the post rigor state is not primarily populated in solution (demonstrated by SAXS experiments) and argue that this state is selected/stabilized by crystal contacts. What is the primary motive to discuss this result with a figure and again in the subsection “Specific recognition of the degenerate IQ motifs in the atypical PfMyoA Lever Arm” in terms of the contacts between W777 and the α5 helices of the PfELC. It is not clear from the manuscript how significant the 'folded-back' position for this study. If this state arises from crystal packing artifacts, are the authors arguing that this state is functionally essential but too transient to capture in solution? Please clarify.

We performed SAXS experiments to evaluate whether the pliant region stays mainly helical promoting converter/ELC interactions as found in the PPS structure for the molecule (see Figure 1D), or whether as seen in the PR structure, it could more easily kink and allow the lever arm to interact with the head. The conclusion is that the pliant region is unlikely to kink in a significant manner. This can be seen as a minor point compared to the rest of what is presented in the paper, but it is also a way to extend our understanding of how stable the Converter/ELC interactions are. This information is important to define the angles explored by the lever arm in each structural states explored during the motor cycle. Whether the pliant region could play a role in an auto-inhibited form of the motor and the characterization on how the Converter/ELC interface could allow the lever arm to uncouple under strain was the primary motive but since SAXS excludes the presence of this form in a significant amount, we do not discuss this further in the manuscript.

2) Prior work (Bookwalter et al., 2017), showed that in the absence of PfELC, PfMyoA-MTIP could move actin filaments at a velocity of ~1.8 µm/s and had an ATPase Vmax of ~130 s^-1^. These data suggest that PfMyoA is still functional without PfELC. However, in this study, a conditional knockout of the Pfelc gene led to a ~complete ablation of red blood cell invasion. Please cite and discuss these data.

We were also surprised that the motor lacking PfELC blocked invasion given that it retained significant function in vitro. We thus performed ensemble force measurements expecting to see a reduction, but the data showed that the force was very similar to the motor with both light chains (new Figure 6—figure supplement 1E). This is unlike what was seen with skeletal muscle myosin lacking the ELC that showed a 2-fold reduction in isometric force (VanBuren et al., 1994). One possibility that we plan to pursue in the future is whether the atypical priming of the PfMyoA lever arm in the PPS state, which results from several interactions that are not present in conventional myosins, is responsible for allowing force to remain high in the absence of the ELC.

It should be noted that a recent pre-print by two of the co-authors (Blake et al., 2020) showed via video microscopy that PfELC cKO merozoites can still deform red blood cells, unlike cKO of PfMyoA heavy chain. Nonetheless, cKO of either PfMyoA or PfELC left merozoites entirely incapable of invasion. The reduced speed of PfMyoA in vitro without PfELC may thus be sufficient to disrupt invasion. Alternative possibilities are that in vivo other factors need to be considered that are not assessed by the in vitro assays. Perhaps in vivo the lack of the PfELC allows proteolysis to occur that would effectively decrease the number of functional motors, or that the shortened neck in the absence of the PfELC does not allow the motor to reach the actin filaments as effectively as the WT does within the constraints of the space between the plasma membrane and inner membrane complex. These possibilities are mentioned in the Discussion at the end of the third paragraph as follows:

“Surprisingly, the ensemble force of PfMyoA lacking the PfELC is comparable to that of the motor with both light chains (Figure 6—figure supplement 1E). […] In addition, the shortened neck in the absence of the PfELC may not allow the motor to reach the actin filaments as effectively as the WT does, within the constraints of the space between the plasma membrane and inner membrane complex.”